# High-Throughput Preparation and High-Throughput Detection of Polymer-Dispersed Liquid Crystals Based on Ink-Jet Printing and Grayscale Value Analysis

**DOI:** 10.3390/molecules28052253

**Published:** 2023-02-28

**Authors:** Rui-Juan Xiong, Yun-Xiao Ren, Yong-Feng Cui, Shu-Feng Cai, Wan-Li He, Xiao-Tao Yuan

**Affiliations:** 1Department of Chemistry and Chemical Engineering, School of Chemistry and Biological Engineering, University of Science and Technology Beijing, Beijing 100083, China; 2Department of Materials Physics and Chemistry, School of Materials Science and Engineering, University of Science and Technology Beijing, Beijing 100083, China

**Keywords:** high-throughput, polymer-dispersed liquid crystal, saturation voltage, grayscale value

## Abstract

In this paper, based on high-throughput technology, polymer dispersed liquid crystals (PDLC) composed of pentaerythritol tetra (2-mercaptoacetic acid) (PETMP), trimethylolpropane triacrylate (TMPTA), and polyethylene glycol diacrylate (PEGD 600) were investigated in detail. A total of 125 PDLC samples with different ratios were quickly prepared using ink-jet printing. Based on the method of machine vision to identify the grayscale level of samples, as far as we know, it is the first time to realize high-throughput detection of the electro-optical performance of PDLC samples, which can quickly screen out the lowest saturation voltage of batch samples. Additionally, we compared the electro-optical test results of manual and high-throughput preparation PDLC samples and discovered that they had very similar electro-optical characteristics and morphologies. This demonstrated the viability of PDLC sample high-throughput preparation and detection, as well as promising application prospects, and significantly increased the efficiency of PDLC sample preparation and detection. The results of this study will contribute to the research and application of PDLC composites in the future.

## 1. Introduction

Polymer-dispersed liquid crystal (PDLC) is a composite thin film material composed of micron-sized liquid crystal (LC) droplets uniformly dispersed in a polymer matrix [1,2]. When no electric field is applied, there is a difference in the refractive index between the LC and the polymer in the positive mode PDLC film. The random distribution of the LC droplet and the director in the system results in the intense scattering of the incident light, so the PDLC presents an opaque opalescent state. When a strong external electric field is applied, the LC molecules in the PDLC film are oriented along the direction of the electric field and match the refractive index of the polymer. The incident light will directly pass through the interface, and the PDLC film switches from opalescent opaque state to transparent state. The optical state of a reverse mode PDLC film before and after driven by an electric field is opposite to that of a positive mode PDLC film [3,4]. Due to its electro-optical properties, PDLC has good future prospects in the fields of smart windows [5], flexible displays [6], holographic gratings [7], and biosensors [8].

Nucleophilic-initiated thiol-ene click reaction is a novel and efficient method for the preparation of PDLC films. PDLC prepared by thiol-ene click reaction has the advantages of simple reaction conditions, high efficiency, and easy access to reaction raw materials [9,10]. Mohsin et al. [11] prepared films with low saturation voltage and medium-high contrast by adjusting the functionality of thiol monomers and crosslinkers in the PDLC system. Ren et al. [12] found that the addition of Capcure 3–800, a 5.5 wt% high molecular weight polymer, as a curing agent, to the thiol-ene reaction system can effectively reduce the saturation voltage of PDLC films. Therefore, thiol-ene click reaction has broad application prospects. This paper intends to carry out research on a click reaction, but the current PDLC research adopts the traditional manual mixing method, and the content of each component needs to be accurately controlled during the mixing process, which is time-consuming, labor-intensive, inefficient, cannot achieve rapid preparation of a large number of samples, and manual mixing will have some man-made uncontrollable errors. The difference between different samples may also be due to external conditions. Therefore, there is an urgent need for an accurate, rapid, and batch preparation of polymer-dispersed LCs.

It is well known that high-throughput methods are becoming more widely used. As early as 1995, researchers [13] screened compounds by establishing a sample combination library to improve the detection rate of materials and the test rate of theoretical predictions. Later, Hansen [14] prepared more than 80 kinds of polymer gradients by inkjet printing, verified the changes of composition in the sample library through analytical methods, and provided a proof of concept for the successful application of polymer gradient arrays as screening tools. At present, high-throughput methods have been widely used to screen catalysts [15] or enzymes [16], solar cells [17] and biological sequencing [18]. In addition, Welch [19] et al. developed a technique for microplate synthesis and screening of new stationary phases, creating a platform for rapid screening of bulk adsorbents, where the analysis speed can be increased by more than a thousand times. Ning [20] reening of up to 1536 reactions on nanomole scales in parallel, and successful reproduction of screening hits at medicinal chemistry-relevant scales. In our preliminary work, we successfully realized the preparation of 1080 BPLC samples based on inkjet printing with high throughput and screened out BPLC materials with a relatively wide temperature range through the high-throughput recognition system of machine learning [21]. However, high-throughput detection of the electro-optical performance of LC samples was not achieved.

So, in this paper, we propose a method based on machine learning to identify the grayscale value to obtain the grayscale-voltage curve and enable quick screening of the saturation voltage value of the sample, to realize the high-throughput detection of PDLC samples. Firstly, PDLC was prepared by the thiol-ene reaction system. Next, 125 sample formulas were designed through comprehensive experiments, and LC samples for high-throughput detection cells were created using the standard curve method. Finally, the driving voltage of all samples and the sample with the lowest driving voltage value were successfully screened out using high-throughput identification of the grayscale value. Based on this method, PDLC can not only quickly prepare multi-batch formulations, and improve research efficiency, but also achieve the purpose of rapid screening of low saturation voltage samples, which can play a role in scientific research and practical applications.

## 2. Results and Discussion

### 2.1. High-Throughput Preparation PDLC Samples

We designed a comprehensive experiment using a three-factor, five-level sample formulation to study the effects of PEGDA600, PETMP, and TMPTA (as shown in Figure 1) on PDLC voltage, and the values of each influencing factor are shown in Table 1. A total of 125 (5 × 5 × 5) combinations can be used for high-throughput preparation of all PDLC samples in a comprehensive design, as detailed in Appendix A.

SLC-1717 (2.4 g) was added into 30 wt% cyclohexanone (5.55 g) solution as the ink of the Y channel of the printing nozzle and PEGDA-600 (1.2 g), PETMP (1.2 g), and TMPTA (0.4 g) were dissolved in cyclohexanone to prepare 15 wt%, 15 wt%, and 5 wt% inks corresponding to the channels of C, M, and K, so as to establish the standard curve of each ink solution [22]. Standard graphs of the four solvents are shown in Appendix A.

A CMYK mode-based pattern of the printed sample library was constructed based on the combined ratio of the standard curve resulting from the configured ink concentration and the comprehensive experimental design. The sample pattern shown in Figure 2a was created by using the CorelDraw program to design a circle with a sample point size of 2 mm and an interval of 3 mm, as well as a filling component allocation ratio. Convert the mass fraction of each sample component into CMYK color parameters. As illustrated in Figure 2b, ink containing SLC-1717, TMPTA, PEGDA-600, and PETMP as solutes is successively injected into each of the four CMYK channels and then placed in a 60 °C oven for 3 h to evaporate the solvent while allowing the solution’s ingredients to properly mix and diffuse. In order to control the thickness of the LC film, 20 μm thick glass beads were equally coated on both sides of the sample after it had been built. The LC cell was then exposed to 365 nm ultraviolet radiation at 10 mW/cm^2^ for 5 min to fully polymerize the monomers. 

The optical state of the high throughput LC samples was similar to that of the conventional PDLC film before and after applied by the external electric field. When no electric field was applied, the non-ideal spherical shape of LC droplets dispersed in the polymer network and were affected by the elastic force and the anchoring force of the network. The LC molecules thus randomly arranged, so there was a strong light scattering at the interface between the LC droplets and the polymer network, and the samples between the orthogonal polarizers showed a bright state of scattering. When a strong external electric field was added, the positive LC molecules were driven by the electric field and arranged vertically on the substrates. The incident light can therefore pass through the interface between the LC droplets and the polymer network, and the samples then appeared in a dark state after extinction in two orthogonal polarizations [3,4]. Figure 2c,d shows photos of the high-throughput preparation of PDLC samples under an electric field and without an electric field. It can be seen from the figure that each sample is clearly visible. Since the composition and total weight of each sample are different, and the LC sample also flows or diffuses before polymerization, the size of the sample points is somewhat inconsistent. However, the size of the sample will not affect the electro-optical performance results of each sample.

### 2.2. High-Throughput Detection of PDLC Sample Voltage

It is well known that when the refractive index of the LC in PDLC does not match the refractive index of the polymer, it appears milky white, and a certain voltage was applied, so that the film can be switched to a transparent state [3,4]. Based on the above changes, according to the principle of electro-optical detection equipment for PDLC, we propose a camera capture method to collect the optical state changes of PDLC samples. That is, the high-throughput technique was chosen based on the corresponding change in grayscale and voltage, which can be obtained by employing PDLC under high-throughput detection equipment.

The electro-optical performance acquisition of samples was carried out in the way shown in Figure 3. The prepared samples were placed on the platform, and the bottom layer was placed with a plane light source, such as a halogen fiber optic light source, which was used as an incident light source and produced a wide area of uniform illumination of flat light with adjustable light intensity. Further, the top high-speed camera (the camera mode with a frame rate of 100 FPS) was utilized to gather the change in the sample before and after the voltage. Orthogonal polarizers were positioned on either side of the sample to enable sensitive detection of the sample’s grayscale value data. In order to power up the sample at a voltage of 60 V, the sample was clamped with a voltage clamp, and the magnitude of the applied voltage was regulated by a voltage amplifier. Through the use of video editing software and image processing software, the video of the change-capturing process was edited, framed, and cropped (for example, Premiere and Photoshop). Since the camera was supported by a stable platform during the collection process, the sample point’s relative position is fixed, allowing us to mark its location with programming software in order to identify the sample point’s serial number and get the related positioning figure. Enter the photo after frame extraction into the LABVIEW visual recognition program, identify and read the grayscale value of the sample according to the positioning figure, choose 100 photos when the grayscale value reaches the lowest value, and normalize the 100 grayscale values. 

It is well known that the saturation voltage of PDLC membrane refers to the voltage applied when the transmittance of PDLC membrane reaches 90% of the maximum transmittance [23,24]. In order to facilitate the high-throughput collection of the electro-optical performance of the sample, we collect the gray value of the PDLC sample during the application of the electric field. It is found that when the sample is placed between the orthogonal polarizers, the influence of other stray light can be eliminated, but this makes the measured gray scale results have the opposite trend with the actual transmittance of the sample, that is, the larger the gray scale value of the sample point, the smaller the transmittance of the actual sample. In order to facilitate the subsequent data analysis, we take the inverse of the gray value of each sample, that is, subtract the gray value of each sample from the maximum gray value (255) and normalize the difference. At this time, the resulted gray value after the value inversion and normalization transformation can be defined as the transmissivity of the PDLC sample, and the maximum value reflects the maximum transmissivity of the PDLC. Therefore, the applied voltage at 90% of the maximum value at this time can be defined as the saturation voltage of the sample. 

### 2.3. Saturation Voltage Analysis of Samples

After high-throughput preparation and detection of PDLC, saturation voltage values were obtained for 125 samples, and the formula and its corresponding saturation voltage values are shown in Appendix A. A four-dimensional diagram as shown in Figure 4 was used to fully comprehend the effect of each component on the saturation voltage of the sample. In this diagram, the LC content is represented by the z-axis, the content of other components is represented by X, and Y axis and the voltage range are expressed by the color and size of the sphere, and the influence of each component on the voltage can be found.

There are similar color and size changes at different PEGDA600 content, so only the figure at its content of 28 wt% is selected as an example, as shown in Figure 4a. The coordinate axis of the four-dimensional figure was the content amount of PETMP, TMPTA, and LC. It can be found from the figure that the voltage has a significant tendency to decrease when the PETMP content remains unchanged, with a decrease in the short-chain TMPTA content and a slight increase in the LC content. This is due to the decrease in the crosslinker content of the short chain, which reduces the functional content of acrylate in the system, resulting in a decrease in the polymerizable monomer in the reaction. Coupled with the increase in LC content, it is easy to form larger liquid crystal droplets, which can reduce the anchoring effect of the polymer network on the liquid crystal [11], so the voltage is reduced. When the TMPTA content of the short-chain crosslinker remains unchanged, the mercaptan monomer content decreases and the LC content also increases significantly. It was observed that the voltage also tends to decrease, which is due to the low functional content of the mercaptan monomer, which cannot provide more active centers, resulting in a slower reaction rate [25] and then a decrease in voltage. When the mercaptan content is 35 wt%, a four-dimensional diagram with PEGDA600, TMPTA, and LC content as the coordinate axis is established, as shown in Figure 4b. When the crosslinker content of the short chain remains constant, as the content of the crosslinker of the long chain increases and the liquid crystal content decreases, the color of the sphere gradually becomes lighter and the volume of the ball is gradually getting smaller, that is, the voltage has a tendency to decrease. This is due to the fact that the increase of long-chain crosslinkers will reduce the number of double bond moles that can be polymerized in the system. The number of free radicals in the polymerization process will also decrease, while the degree of polymerization crosslinking will decrease [26] and the voltage will also decrease in this trend. When the TMPTA content of the short-chain crosslinker is 7 wt%, it can be found from Figure 4c that the long-chain crosslinker PEGDA600 content is certain. With the decrease in the functional content of the mercaptan monomer and the large increase in LC content, the density of the polymer network will be reduced [25] and comprehensive consideration will lead to a decrease in voltage. When the mercaptan content is fixed, with the long-chain crosslinker content decreasing, the LC content increases slightly, which will make the voltage tend to increase, mainly caused by the decrease in the content of polymerizable monomers in the system. 

Therefore, taking into account the influence of various factors, combined with the four-dimensional diagram shown in Figure 4d with the functional degree of thiol monomer, acrylate functionality, and LC doping as coordinates, it can be found that PDLC is mainly affected by the functional degree of the mercaptan monomer and the functional content of acrylate in the system. When more mercaptan monomer functional groups dope, the higher the voltage. Similarly, with more doped acrylate functional groups, the voltage also tends to increase. Therefore, from this experiment, it can be known that when the molar ratio of thiol-olefinic functional group is 1:0.67, the driving voltage of PDLC is the smallest.

Figure 5a shows the average voltage scatterplot of PEGDA600 at 12 wt%, 16 wt%, 20 wt%, 24 wt%, and 28 wt%, respectively. From the figure, it can be seen that when the content of crosslinker PEGDA600 gradually increases, the voltage decreases with the increase, which is due to the increase of the crosslinker of the chain length to increase the molecular weight, leaving a smaller number of molecules under the same mass ratio of each sample, which leads to a decrease in the crosslinking point, a decrease in the degree of crosslinking of polymerization, and a decrease in voltage [27]. Therefore, the optimal content of PEGDA-600 was determined to be 28 wt%. Figure 5b shows the effect of the content of the crosslinker TMPTA on the saturation voltage. It can be seen from the figure that with the increase of the crosslinker content of the short chain, the increase of acrylate functionality in the system will accelerate the reaction speed, which has a significant impact on the tightness of the polymer network, and the saturation voltage has a significant tendency to increase [27], so it can be determined that the optimal content of the crosslinker TMPTA is about 3 wt%. Figure 5c shows the line chart of PETMP content and voltage, since the functional degree of the mercaptan monomer plays the role of curing agent, with the increase of thiol functional content, the curing reaction speed and crosslinking speed also increases, more active centers are displayed, and the polymerization rate is increased [28]. When there is an excess of mercaptan polymerizable monomers, a denser polymer network structure is formed, resulting in an increase in saturation voltage. Therefore, in combination with the line chart, it can be determined that the optimal PETMP content in this formulation is 20 wt%.

Therefore, considering the influence of various factors on the driving voltage, using the high-throughput detection platform, it was learned that the saturation voltage of sample No. 20 was as low as 16.8 v, and its proportions were composed of pentaerythritol tetra(2-mercaptoacetic acid) (PETMP), trimethylolpropane triacrylate (TMPTA), polyethylene glycol diacrylate (PEGD 600), and liquid crystal (SLC1717) as 20 wt%/3 wt%/28 wt%/49 wt%. 

### 2.4. Comparative Experiments of High-Throughput Preparation versus Manual Preparation 

To verify the accuracy of the electro-optical performance of the LC high-throughput detection cell, we randomly selected the first 20 samples shown in Appendix A for manual mixing, which was very close to the saturation voltage values of the high-throughput prepared samples, as shown in the voltage value comparison chart shown in Figure 6a. At the same time, we manually mixed sample No. 20 with sample No. 20, measured using a high-throughput detection platform for a voltage-transmittance curve, as shown in Figure 6b. From the figure, we know that the curves of manual mixing samples and high-throughput preparation samples were similar, and the saturation voltage values of manual mixing and high-throughput preparation samples are 16.97 V and 16.8 V, respectively, and there may be certain deviations from the numerical and curve points. These errors are mainly caused by the following two reasons: On the one hand, because the thickness of the LC cell will also affect the saturation voltage of the PDLC film, the glass substrate was too large and may make different parts of the LC cell have a certain thickness unevenness, and may be due to the brightness of the plane light source not being uniform enough so that the collected photos are distorted [21]. On the other hand, due to the different test conditions of the two, that is, the manually mixed PDLC film material is normalized with an air transmittance of 100%, and inkjet printing was directly normalized by the grayscale value, so there will still be some differences in the curve from the coordinate axis. Further, it may also be because of some manual errors during manual mixing, resulting in the existence of saturation voltage errors between the saturation voltage of each sample point and the saturation voltage of a single sample mixed manually, but it can be seen that the overall trend is consistent, and the accuracy of the saturation voltage of high-throughput detection can also be seen from this aspect. At the same time, the morphology of sample No. 20 was photographed by scanning electron microscopy, as shown in Figure 6c,d, showing the topography of PDLCs prepared by manual mixing and high-throughput, and it can be found that the two have very similar morphological structures.

From the above, it can be found that although the high-throughput method may have some deviations in the saturation voltage of the detection PDLC, these deviations are still within the acceptable range. This method cannot only perform large-area high-throughput detection but also all samples are tested under one experimental condition, which can greatly avoid accidental errors caused by changes in external conditions. Additionally, to a certain extent, the research efficiency of PDLC is improved, which is more economical and efficient than traditional detection devices and conducive to large-scale material screening research.

## 3. Materials and Equipment

The nematic LC SLC1717(T_NI_ = 365 K, Δn = 0.201, n_o_ = 1.519) used in this work was purchased from Yongsheng Huaqing Liquid Crystal Co., Ltd., (Shijiazhuang, China). Thiol monomer Pentaerythritol tetrakis(3-mercaptopropionate) (PETMP) was purchased from Xiensi Biochemical Technology Co., Ltd., (Tianjin, China). The flexible chain crosslinking agent polyethylene glycol diacrylate (PEGDA600) was purchased from Libai Biotechnology Co., Ltd., (Shanghai, China). The multifunctional crosslinking agent trimethylolpropane triacrylate (TMPTA) was obtained from Aladdin Industries (Beijing, China). The photoinitiator Irgacure 651 was purchased from Hongbai Technology Co., Ltd., (Beijing, China). The chemical structures of all materials, shown in Figure 1, were commercially available and used without further purification.

A piezoelectric drop-on-demand (DOD) inkjet printer (OS-A3UV-05, China Shenzhen Dongsheng Co., Ltd., Shenzhen, China) was used to print high-throughput detection cell samples, each taken by a Sony camera (version 2.43, Cao Yingwei, Irvine, CA, USA). Individual PDLC samples were tested for electro-optical properties at room temperature using a liquid crystal device parameter tester (LCT-5016 C, China Changchun Liancheng Instrument Co., Ltd., Jilin, China). The structure of the device for high-throughput detection of LC samples is shown in Figure 3 in Section 2.2. The programming software used was mainly Labview and functional modules related to image and data processing. The polymer morphology of PDLC films was observed by scanning electron microscopy (SEM, Carl Zeiss, Oberkochen, Germany, GeminiSEM 300). The LC samples were immersed into cyclohexane at room temperature for approximately 2 weeks to remove LC from the polymer networks, followed by vacuum drying at 80 °C for 24 h.

## 4. Conclusions

High-throughput preparation of PDLC was realized based on inkjet printing, and high-throughput detection of the electro-optical performance of samples was realized by machine vision using the grayscale method. That is, 125 LC samples were designed through high-throughput preparation and screening, and the effects of the functionality of thiol monomers and crosslinker functionality on the electro-optical properties of PDLC films were studied by using high-throughput detection equipment and batch reading procedures. Sample No. 20 was successfully screened with a minimum saturation voltage of 16.8 v, and the ratio of each component PETMP/TMPTA/PEGDA600/LC was 20 wt%/3 wt%/28 wt%/49 wt%. In addition, we also found that the electro-optical properties and morphology of high-throughput prepared samples and manual mixing samples were very similar. Although there are some errors in high-throughput preparation and manual mixing, overall, the similarity between voltage-grayscale values and actual voltage transmittance indicates the feasibility of high-throughput preparation and detection. This experimental method can select the optimal formula from a large number of PDLC samples so that the efficiency is greatly improved, the equipment structure is simple, the operation is convenient, and it can achieve rapid batch preparation. It is expected that PDLC dimming films with the best electro-optical performance will be prepared according to such methods in the future.

## Figures and Tables

**Figure 1 molecules-28-02253-f001:**
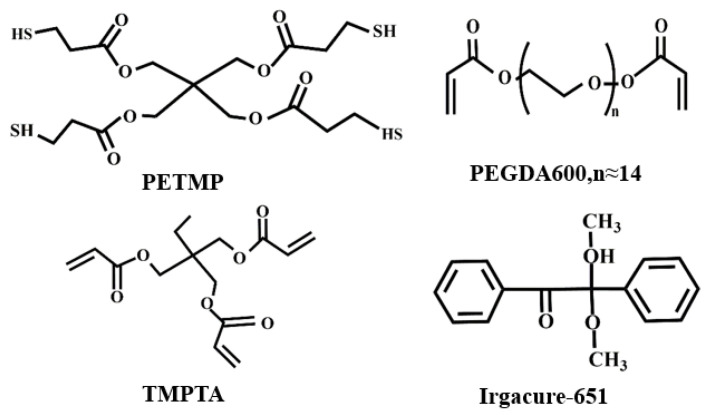
Chemical structures of monomers used in this study.

**Figure 2 molecules-28-02253-f002:**
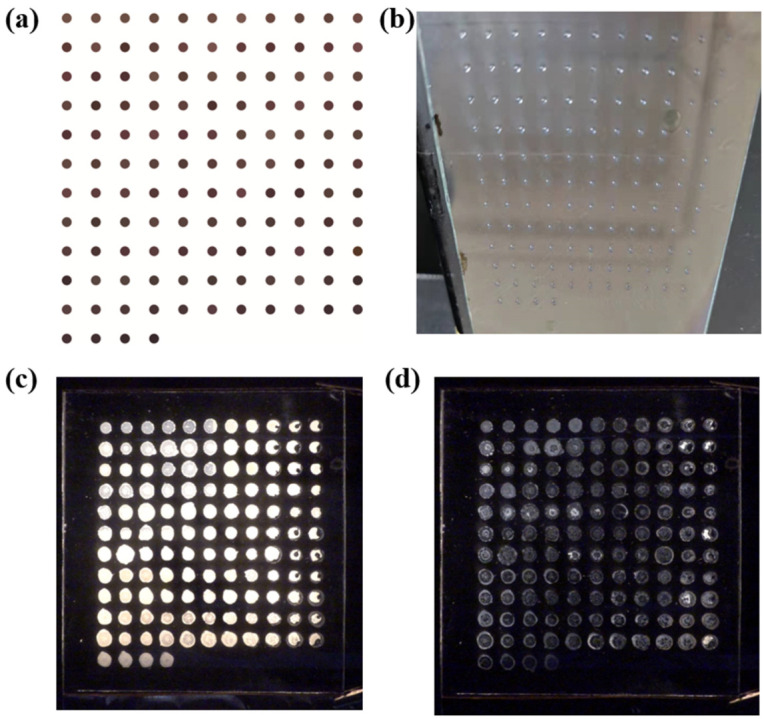
(**a**) Color fill pattern corresponding to 125 sample points based on CMYK mode; (**b**) Photographs of 125 samples obtained by inkjet printing; (**c**) Photographs of 125 samples between crossed polarizers without electric field; (**d**) Photographs of 125 samples between crossed polarizers under electric field.

**Figure 3 molecules-28-02253-f003:**
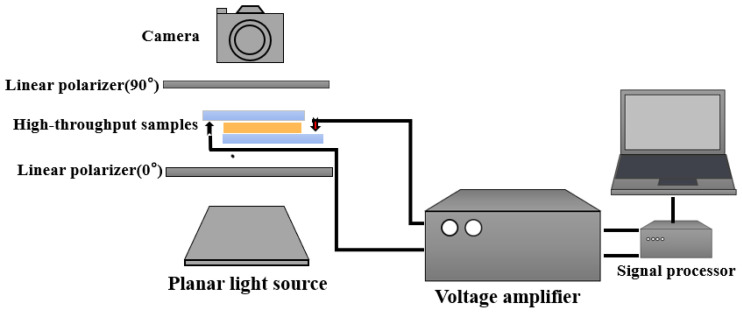
Device for high-throughput detection of LC samples.

**Figure 4 molecules-28-02253-f004:**
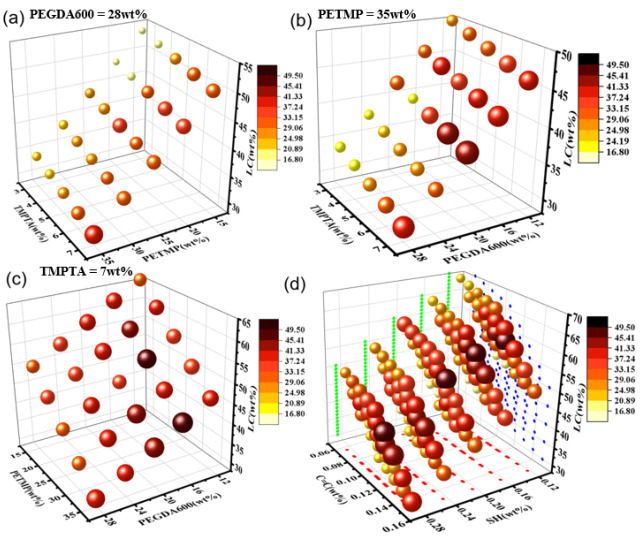
(**a**) Four-dimensional diagram with PETMP, TMPTA, and LC content as the coordinate axis; (**b**) Four-dimensional diagram with PEGDA600, TMPTA, and LC content as the coordinate axis; (**c**) Four-dimensional diagram with PEGDA600, PETMP and LC content as the coordinate axis; (**d**) Four-dimensional diagram with the functionality of thiol monomers, crosslinker functionality and LC content as coordinates.

**Figure 5 molecules-28-02253-f005:**
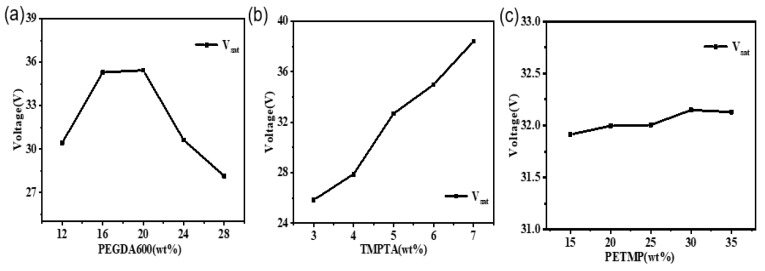
Saturation voltage for samples with different content: (**a**) PEGDA-600; (**b**) TMPTA; (**c**) PETMP. The data were obtained using orthogonal analysis software (Design-Expert 8.0). The target formation in the figure was not affected by the composition of other components, that is, the composition characteristics do not change with the increase of voltage.

**Figure 6 molecules-28-02253-f006:**
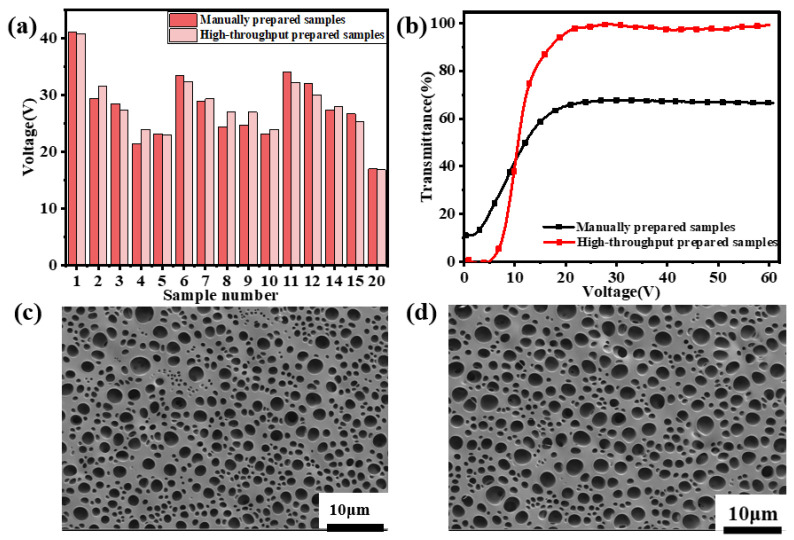
(**a**) Comparison of the saturation voltage of the sample prepared by manual mixing and high-throughput preparation; (**b**) Voltage-transmittance curve of sample No. 20 under manual mixing and high-throughput preparation; (**c**) Micrographography for the preparation of PDLC morphology by high-throughput preparation (sample No. 20); (**d**) Microphotography for manual mixing to prepare PDLC morphology (sample No. 20).

**Table 1 molecules-28-02253-t001:** Factor value and level value of the experimental design.

	Factor Value	PEGDA-600 (wt%)	PETMP (wt%)	TMPTA (wt%)
Level Value				
1		28	35	7
2		24	30	6
3		20	25	5
4		16	20	4
5		12	15	3

## Data Availability

Not applicable.

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
