# Peer review of "High-Throughput Preparation and High-Throughput Detection of Polymer-Dispersed Liquid Crystals Based on Ink-Jet Printing and Grayscale Value Analysis"

_molecules, 2023, doi:10.3390/molecules28052253_

Round 1

Reviewer 1 Report

The paper reports high-throughput preparation and analysis of PDLC samples. This approach allowed for time- and material-efficient optimization of PDLC. The technique is important and the results look promising, however, there are some comments and questions to the authors, especially regarding the design of experiment:

1.     The authors should reason the choice of the system, which they optimize: why certain cross-linkers were selected, what the functional differences between them are, and what defined the concentration range and ratios of the components selected. Also, as the comprehensive analysis of the experimental data has shown that the optimal content of the both cross-linkers are at the very edges of the ranges studied, this content may be not optimal and it looks that further increase of PEGDA content or decrease of PETMP content (even down to 0) could further improve the electro-optical response.

2.     The PDLC samples non-uniformity and depolarization (fig. 2c,d) should be commented.

3.     It is quite challenging to produce such a large area cell with a uniform thickness. As the thickness is critical for the driving voltage, how did the authors provide and control its uniformity during the experiment, and if they didn’t, how do they know that the thickness variation didn’t interfere with the trends observed? 

4.     There are examples of unclear language throughout the text. I suggest that English should be checked and improved in order to provide adequate reading. Typos which also present should be corrected

Minor corrections:

1.     Ref 19 – reference in the “Reference” section doesn’t match the text, p.2: … and Chris Buddha [19] et al. have developed …

2.     Fig. 4 – the fixed values of PEGDA for 4a, PETMP for 4b, and TMPTA for 4c should be denoted in the figure caption to facilitate its reading

3.      Axes scales for Fig. 4d should be brought to correspondence with Figs. 4a-c or it should be denoted how the values correspond to each other.

Reviewer 2 Report

The manuscript present an approach for high-throughput preparation of polymer-dispersed liquid crystals (PDLCs) and characterizes how the composition and method of preparation affect parameters such as switching voltage. The description of the technology is fine, but I do not find any scientific insight into the electro-optical properties of the composition. It is known, for example, that the switching of the PDLCs depends strongly on the type of surface anchoring of the director: Homeotropic anchoring produces defects-hedgehogs in the droplets while tangential anchoring produces two point defects-boojums at the poles. When the field is applied, the structures respond differently. However, the manuscript present no data on what is the director orientation at the nematic-polymer interface and what might be the structure of the droplets. It might be a good idea to add experiments on larger droplets, around 5-8 micrometers in diameter, to get the idea about surface anchoring and thus the switching mechanism. Figure 5 could also be improved: It shows how the saturation voltage depends on the composition, but one would also need to see how the properties of compositions change as the voltage increases and at which point the system "saturates" or reaches 90% of transparency, which, according to the authors, defines the saturation voltage.

Round 2

Reviewer 1 Report

The authors have addressed most of my concerns. There is still room for English improvement, which I leave to editors. Regarding the rest, I am pleased to recommend this paper for being published in Molecules

Author Response

Dear Editors and Reviewers,
Thank you for your letter and the reviewers' comments on our manuscript titled "High-Throughput Preparation and High-Throughput Detection."
Polymer dispersion liquid crystals based on inkjet printing and grayscale value analysis (ID: crystal-2155521). These opinions are valuable for the revision and improvement of our papers, and have important guiding significance for our research. We are very happy to receive your approval, and we will strive to improve the quality of the manuscript

Reviewer 2 Report

I am disappointed by how the authors modified their manuscript.

1.     Instead of addressing the issue I raised in the first review (of the surface anchoring of the director at the liquid crystal-polymer interface), they added a badly written passages on light scattering at the liquid crystal droplets and dielectric anisotropy. They added a figure to their reply to the reviewer (in which the anchoring is illustrated as tangential) but did not offer any explanation nor a figure to the revised text. 

2.     The scientific presentation and English in newly added pieces are not satisfactory (statements such as  “birefringence of light” , “dielectric positive”, “vertically oriented to the substrate direction” must be corrected). The statement “the application of gravity to prepare LC cells is not uniform” would puzzle any reader who would come across this text. There are other examples of bad English/scientific formulations that must be corrected.

3.     The definition of the saturation voltage on page 5 is not satisfactory.

4.     It is not clear in what sense the diagrams are “four-dimensional” in Fig.4

5.     It is not clear which “figure” the authors refer to in the newly inserted text on page 4.

I regret to conclude that the revision did not improve the manuscript, it made it worse. 

Author Response

Dear Editors and Reviewers:

Thank you for your letter and for the reviewers’ comments concerning our manuscript entitled “High-throughput Preparation and High-throughput Detection of 
Polymer-dispersed Liquid Crystals Based on Ink-jet Printing and Grayscale Value Analysis” (ID: crystals- 2155521). Those comments are all valuable and very helpful for revising and improving our paper, as well as the important guiding significance to our researches. We have studied comments carefully and have made correction which we hope meet with approval. The modified part is marked with the “Track Changes” function in the paper. The main corrections in the paper and the responds to the reviewer’s comments are as flowing:

1.Instead of addressing the issue I raised in the first review (of the surface anchoring of the director at the liquid crystal-polymer interface), they added a badly written passages on light scattering at the liquid crystal droplets and dielectric anisotropy. They added a figure to their reply to the reviewer (in which the anchoring is illustrated as tangential) but did not offer any explanation nor a figure to the revised text. 

Respond: In this paper, we study the different proportions of PDLC prepared by high-throughput preparation of inkjet printing to study the saturation voltage of the sample. Then, as for the orientation of liquid crystals and the various states of anchoring, it is not a scope of our study in this article. We are going to consider this part in the next article, where we will only consider the effect of this component on saturation voltage

  1. The scientific presentation and English in newly added pieces are not satisfactory (statements such as  “birefringence of light” , “dielectric positive”, “vertically oriented to the substrate direction” must be corrected). The statement “the application of gravity to prepare LC cells is not uniform” would puzzle any reader who would come across this text. There are other examples of bad English/scientific formulations that must be corrected.

Respond: This part in the manuscript has been modified accordingly.

  1. The definition of the saturation voltage on page 5 is not satisfactory.

Respond: This sentence in the manuscript has been modified accordingly.

  1. It is not clear in what sense the diagrams are “four-dimensional” in Fig.4

Respond: Using three-dimensional coordinate axis, color and size are used together to represent the regular change of saturation voltage value, that is, four-dimensional.

5.It is not clear which “figure” the authors refer to in the newly inserted text on page 4.

Respond: The newly inserted text on the 4 page is a description of the PDLC switching mechanism, which is intended to explain the electrical switching mechanism of PDLCs prepared at high throughput, thereby describing the changes of PDLCs under the electric field shown in Figure 2(c) and (d).

Round 3

Reviewer 2 Report

The revised manuscript is improved to the level that I can recommend publication